# Reconciling isothermal titration calorimetry analyses of interactions between complexin and truncated SNARE complexes

Eric A Prinslow[1,2,3], Chad A Brautigam[1,4], Josep Rizo[1,2,3]*

[1]Department of Biophysics, University of Texas Southwestern Medical Center, Dallas, United States; [2]Department of Biochemistry, University of Texas Southwestern Medical Center, Dallas, United States; [3]Department of Pharmacology, University of Texas Southwestern Medical Center, Dallas, United States; [4]Department of Microbiology, University of Texas Southwestern Medical Center, Dallas, United States

**Abstract** Neurotransmitter release depends on the SNARE complex formed by syntaxin-1, synaptobrevin and SNAP-25, as well as on complexins, which bind to the SNARE complex and play active and inhibitory roles. A crystal structure of a Complexin-I fragment bearing a so-called 'superclamp' mutation bound to a truncated SNARE complex lacking the C-terminus of the synaptobrevin SNARE motif (SNARE$\Delta$60) suggested that an 'accessory' $\alpha$-helix of Complexin-I inhibits release by inserting into the C-terminus of the SNARE complex. Previously, isothermal titration calorimetry (ITC) experiments performed in different laboratories yielded apparently discrepant results in support or against the existence of such binding mode in solution (Trimbuch et al., 2014; Krishnakumar et al., 2015). Here, ITC experiments performed to solve these discrepancies now show that the region containing the Complexin-I accessory helix and preceding N-terminal sequences does interact with SNARE$\Delta$60, but the interaction requires the polybasic juxtamembrane region of syntaxin-1 and is not affected by the superclamp mutation within the experimental error of these experiments.
DOI: https://doi.org/10.7554/eLife.30286.001

*For correspondence:
Jose.Rizo-Rey@UTSouthwestern.edu

**Competing interests:** The authors declare that no competing interests exist.

## Introduction

The release of neurotransmitters by $Ca^{2+}$-triggered synaptic vesicle exocytosis is governed by a sophisticated protein machinery that includes the neuronal soluble N-ethylmaleimide sensitive factor attachment protein receptors (SNAREs) synaptobrevin, syntaxin-1 and SNAP-25 as central components (*Südhof and Rothman, 2009*; *Rizo and Xu, 2015*). These proteins form a tight SNARE complex that consists of a four-helix bundle and plays a key role in membrane fusion by bringing the synaptic vesicle and plasma membranes together (*Söllner et al., 1993*; *Sutton et al., 1998*; *Weber et al., 1998*). The exquisite regulation of release also depends on multiple specialized proteins, including Complexins among others. These small soluble proteins bind tightly to the SNARE complex (*McMahon et al., 1995*) and play both active and inhibitory roles in release (*Reim et al., 2001*; *Huntwork and Littleton, 2007*; *Hobson et al., 2011*; *Martin et al., 2011*), but the underlying mechanisms remain unclear.

A crystal structure of the SNARE complex bound to a fragment spanning residues 26–83 of Complexin-I [CpxI(26-83)] showed that binding involves a central $\alpha$-helix of CpxI, while a preceding accessory $\alpha$-helix does not contact the SNAREs (*Figure 1A,B*). Electrophysiological studies indicated that

the accessory helix mediates at least in part the inhibitory role of CpxI, leading to a model whereby the accessory helix inhibits release by replacing part of the synaptobrevin SNARE motif in a partially assembled SNARE complex, thus preventing C-terminal assembly of the complex (*Xue et al., 2007*; *Maximov et al., 2009*). Cell-cell fusion assays supported this model and led to the design of several CpxI mutants with increased or decreased inhibitory activity in these assays, including a 'superclamp' mutant where three charged residues were replaced with hydrophobic residues (D27L, E34F, R37A) to enhance the putative binding to the partially assembled SNARE complex (*Giraudo et al., 2009*).

A crystal structure of a SNARE complex with synaptobrevin truncated at residue 60 (SNAREΔ60) bound to CpxI(26-83) bearing the superclamp mutation [scCpxI(26-83)] later revealed a zig-zag array where the central helix binds to one SNAREΔ60 complex and the accessory helix binds to another SNAREΔ60 complex (*Figure 1C*), suggesting that such an array inhibits neurotransmitter release before $Ca^{2+}$ influx (*Kümmel et al., 2011*). The validity of the scCpxI accessory helix-SNAREΔ60 interaction observed in the structure was supported by isothermal titration calorimetry (ITC) results that we discuss in detail below (*Kümmel et al., 2011*). However, no interaction between the accessory helix of WT CpxI(26-83) or scCpxI(26-83) with C-terminally truncated SNARE complexes was observed by analogous ITC experiments and extensive NMR analyses in a separate study (*Trimbuch et al., 2014*). Moreover, electrophysiological experiments performed in the same study did not detect significant functional effects for the superclamp mutation in CpxI, and led to a model whereby the accessory helix inhibits release because it causes electrostatic and/or steric hindrance with the membranes at the site of fusion (*Figure 1D*) (*Trimbuch et al., 2014*). Note that, in a previous study, the superclamp CpxI mutant was claimed to inhibit spontaneous release more efficiently than WT CpxI (*Yang et al., 2010*), but the data were not inconsistent with the results of *Trimbuch et al. (2014)*. Rescue assays with mammalian CpxI in *Drosophila* Complexin nulls did reveal a stronger inhibition of spontaneous release for superclamp CpxI than for WT CpxI (*Cho et al., 2014*), supporting the hydrophobic interaction observed in the crystal structure of *Kümmel et al. (2011)*. Conversely, the finding that the accessory helix can be functionally replaced by an unrelated, uncharged α-helix in *C. elegans* supported the notion that the inhibitory role of this helix does not involve protein-protein interactions (*Radoff et al., 2014*), suggesting that steric hindrance with the membranes may be sufficient for this role.

The above results and other studies have led to considerably different views on the available data and the merits of the proposed models, which is natural in ongoing investigations of a highly complex molecular mechanism that is still poorly understood. However, it was worrisome and confusing to the field that different results were obtained in the Rothman and Rizo laboratories in ITC experiments that presumably were performed under analogous conditions with the same protein sequences (*Kümmel et al., 2011*; *Trimbuch et al., 2014*; *Krishnakumar et al., 2015*). Here we describe our efforts to identify the source of the discrepancies and present new data showing that there is indeed an interaction between SNAREΔ60 and residues 1–47 of CpxI, although this interaction is not affected by the superclamp mutation in CpxI and requires the polybasic juxtamembrane region of syntaxin-1.

## Results and discussion

The ITC experiments that yielded apparently discrepant results involved blocking assays where SNAREΔ60 was saturated with a CpxI fragment lacking the accessory helix and the mixture was titrated with full-length CpxI or CpxI(26-83), both of which contain the accessory helix. The observation of heat release in the assays performed with WT CpxI, and of an increase in the heat release when the superclamp mutation was introduced in the accessory helix, supported the notion that the interaction of the CpxI accessory helix with SNAREΔ60 observed in the zigzag crystal structure occurs in solution (*Kümmel et al., 2011*; *Krishnakumar et al., 2015*). These conclusions relied on the assumption that the excess of the blocking CpxI fragment used in the competition assays [CpxI(48-134)] completely saturates the SNAREΔ60 complex, which was supported by direct titrations of SNAREΔ60 with CpxI(48-134) that yielded a $K_D$ of 457 nM (*Krishnakumar et al., 2015*). However, another study that used similar conditions to those described in *Kümmel et al. (2011)*, blocking SNAREΔ60 with a 1.5-fold excess of a CpxI(47-134) fragment, observed similar heat release upon

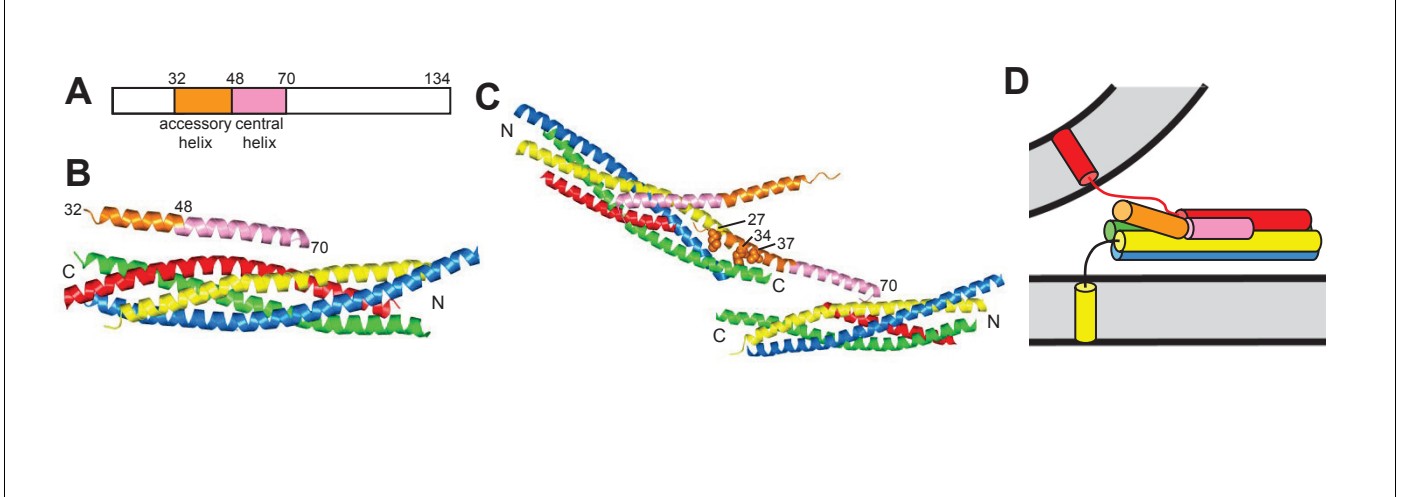

**Figure 1.** Models of the inhibitory function of Complexin. (**A**) Domain diagram of CpxI. Selected residue numbers are indicated above the diagram. (**B**) Ribbon diagram of the crystal structure of the SNARE complex bound to CpxI(26-83) (PDB code 1KIL) (***Chen et al., 2002***). Synaptobrevin is colored in red, syntaxin-1 in yellow, SNAP-25 in blue and green (N-terminal and C-terminal SNARE motifs, respectively), and CpxI(26-83) in orange (accessory helix) and pink (central helix). N and C indicate the N- and C-termini of the SNARE motifs. Selected residue numbers of CpxI(26-83) are indicated. (**C**) Ribbon diagram of the crystal structure of the SNAREΔ60 complex bound to the CpxI(26-83) superclamp mutant (PDB code 3RK3) (***Kümmel et al., 2011***). Two complexes are shown to illustrate the zigzag array present in the crystals. Selected residue numbers are indicated for one of the scCpxI(26-83) molecules, which binds to one SNAREΔ60 complex through the central helix and to another SNAREΔ60 complex through the accessory helix. The three mutated residues in the accessory helix are shown as spheres and their residue numbers are indicated. (**D**) Model postulating that the Complexin accessory helix inhibits neurotransmitter release because of steric repulsion with the vesicle membrane. The model is based on the crystal structure shown in (**A**), but assumes that the C-terminus of the synaptobrevin SNARE motif is not assembled into the SNARE complex. This figure is based on Figure 1 of ***Trimbuch et al. (2014)***, with modifications.

DOI: https://doi.org/10.7554/eLife.30286.002

titration with WT or superclamp mutant CpxI(26-83) (***Trimbuch et al., 2014***). Because direct titrations of SNAREΔ60 with CpxI(47-134) yielded a $K_D$ of 2.4 µM, this study concluded that the heat release observed in the blocking assays arises from incomplete saturation of SNAREΔ60 by CpxI(47-134) rather than from an interaction of the accessory helix with SNAREΔ60. Note that the relatively weak affinity reflected by this $K_D$ is not surprising because the truncation of synaptobrevin in SNAREΔ60 removes multiple residues that contact the central CpxI helix in the crystal structure of CpxI(26-83) bound to the SNARE complex (***Chen et al., 2002***).

To elucidate the reasons for these discrepancies, the Rothman laboratory shared the expression vectors used in ***Kümmel et al. (2011)*** and ***Krishnakumar et al. (2015)*** with the Rizo laboratory, so that we could rule out the possibility that the distinct results obtained in ***Trimbuch et al. (2014)*** arose from differences in the protein fragments used. In addition, E. Prinslow from the Rizo laboratory visited the Rothman laboratory. In the resulting discussions, the Rothman laboratory explained an experimental detail that had not been reported in ***Kümmel et al. (2011)*** and ***Krishnakumar et al. (2015)***: in the blocking assays monitored by ITC, sufficient excess of CpxI fragment [CpxI(48-134)] to block SNAREΔ60 was added so that minimal heat release was observed in control experiments where blocked SNAREΔ60 was titrated with CpxI(48-134) itself. Hence, the (larger) heat release observed in the titrations with CpxI or scCpxI could not arise from incomplete saturation of SNAREΔ60 by CpxI(48-134). Moreover, the conversations between the two laboratories and protein analyses by SDS PAGE revealed that there were differences in the syntaxin-1 fragments used to assemble SNAREΔ60. The Rizo laboratory used a fragment spanning residues 191–253 of syntaxin-1 (***Trimbuch et al., 2014***), as reported in ***Kümmel et al. (2011)*** and ***Krishnakumar et al. (2015)***. However, the Rothman laboratory explained that, for the ITC experiments reported in these two papers, SNAREΔ60 complexes were actually formed with syntaxin-1 fragments spanning residues 188–259 or 188–265, and containing an N-terminal His$_6$-tag. Note that syntaxin-1(191–253) spans most of the SNARE motif except for a few C-terminal residues, which helps to improve the solubility of SNARE complexes (***Chen et al., 2002***), syntaxin-1(188–259) includes the entire SNARE

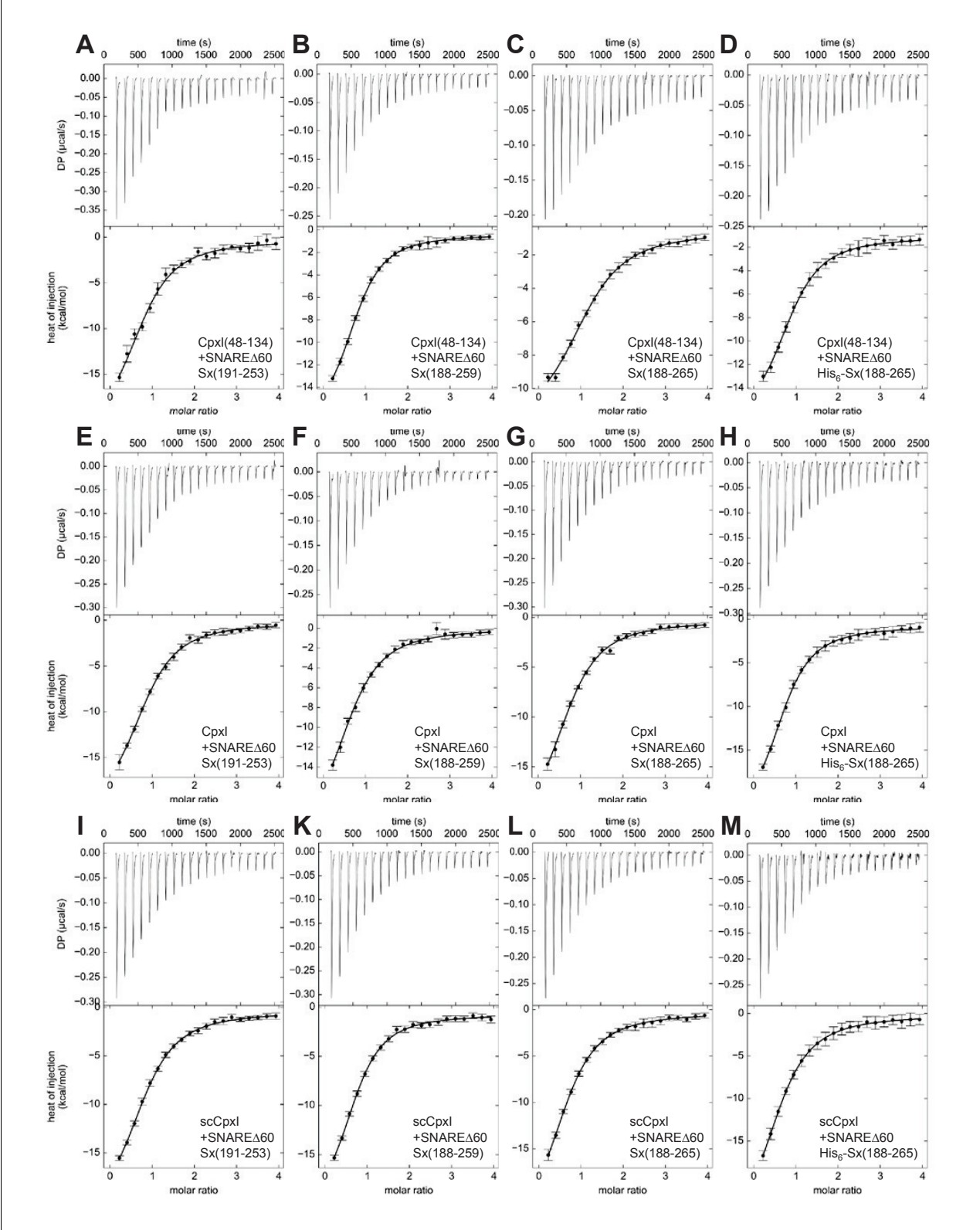

**Figure 2.** ITC analysis of CpxI-SNAREΔ60 interactions by direct titration. The various panels show direct titrations of SNAREΔ60 containing syntaxin-1 (191–253) (**A,E,I**), syntaxin-1(188–259) (**B,F,K**), syntaxin-1(188–265) (**C,G,L**) or His$_6$-syntaxin-1(188–265) (**D,H,M**) with CpxI(48-134) (**A–D**), CpxI (**E–H**) or scCpxI (**I–M**), monitored by ITC. The upper panels show the baseline- and singular-value-decomposition-corrected thermograms for the respective

*Figure 2 continued on next page*

*Figure 2 continued*

experiments. The circles in the lower panels are the integrated heats of injection, with the error bars representing estimated errors for these values (*Keller et al., 2012*). The lines in these panels represent the respective fits of the data to a single binding site 'A + B <->AB' model.

DOI: https://doi.org/10.7554/eLife.30286.003

motif, and syntaxin-1(188–265) contains in addition the juxtamembrane region of syntaxin-1, which includes five positively charged residues.

To investigate how the differences in the syntaxin-1 fragments might affect the ITC results, we performed a systematic analysis using SNAREΔ60 containing syntaxin-1(191–253), syntaxin-1(188–259) or syntaxin-1(188–265) with or without a His$_6$-tag (below referred to as SNAREΔ60-Sx253, SNAREΔ60-Sx259, SNAREΔ60-Sx265 or His$_6$-SNAREΔ60-Sx265, respectively). The analysis involved direct titrations of the various SNAREΔ60 complexes with CpxI(48-134), full-length CpxI or full-length scCpxI mutant, and blocking assays using these complexes. All proteins were expressed using vectors provided by the Rothman laboratory. Representative data are shown in *Figures 2* and *3*, and *Table 1* describes the K$_D$s measured in the direct titrations. All direct titrations performed with the 12 different combinations of SNAREΔ60 complexes and CpxI proteins yielded K$_D$ values around 2 μM, with no marked differences considering the confidence intervals of the measurements. The consistency of these results underlines the reliability of the data and shows that the affinity of CpxI for SNAREΔ60 is not substantially altered by the presence of residues 1–47 of CpxI or by the differences in the syntaxin-1 fragments used to assemble SNAREΔ60.

The systematic blocking assays were performed using the approach designed by the Rothman laboratory, blocking the various SNAREΔ60 complexes with a large (4.9-fold) excess of CpxI(48-134), and titrating with CpxI(48-134) itself, CpxI or scCpxI. Because the use of different total protein concentrations might have yielded some variability in the heat release observed in the previously published blocking assays, all experiments of this systematic analysis used similar total protein concentrations. In all the control experiments where the blocked SNAREΔ60 complexes were titrated with CpxI(48-134) itself, only a very small amount of heat release was observed (*Figure 3A–D*), which can be attributed to a small amount of remaining free SNAREΔ60. Assuming a K$_D$ of 2 μM, this small amount is estimated to be about 2.5% of the total SNAREΔ60 complex, which is consistent with the small heat release observed. Comparable, very small heat release was observed in experiments where blocked SNAREΔ60-Sx253 or SNAREΔ60-Sx259 complexes were titrated with full-length CpxI or scCpxI (*Figure 3E,F,I,K*), indicating that there is no interaction of residues 1–47 of CpxI with these complexes. However, the heat release was higher when full-length CpxI or scCpxI were titrated into blocked SNAREΔ60-Sx265 (*Figure 3G,L*), showing that residues 1–47 of CpxI do interact with SNAREΔ60 when the complex includes the juxtamembrane region in syntaxin-1. Reliable K$_D$s cannot be derived from these data because of the difficulty in accurately defining the baselines in the respective isotherms, but it appears that the interaction is weak based on the small amount of heat release (*Figure 3G,L*) and the fact that the presence of the juxtamembrane region did not lead to an overt increase in the measured affinities in the direct titrations (*Figure 2*, *Table 1*). We also note that even higher heat release was observed in blocking experiments where His$_6$-SNAREΔ60-Sx265 was titrated with CpxI or scCpxI (*Figure 3H,M*), showing that the His$_6$-tag can alter the results and hence should be removed.

Overall, these results show that there is an interaction between the C-terminus of SNAREΔ60-Sx265 and residues 1–47 of CpxI, although the nature of the interaction remains unclear. It seems highly unlikely that the CpxI accessory helix-SNAREΔ60 interaction observed in the zigzag crystal structure (*Figure 1C*) underlies the heat release observed in the blocking assays performed with the SNAREΔ60-Sx265 and His$_6$-SNAREΔ60-Sx265 complexes because the heat release was not markedly altered by the superclamp mutation (*Figure 3G,H,L,M*; see also the superposition of data obtained for WT CpxI and scCpxI shown in *Figure 3—figure supplement 1*). Note that the superclamp mutation replaces three charged residues of WT CpxI with hydrophobic side chains that in the zigzag crystal structure pack against the hydrophobic groove left in SNAREΔ60 by the synaptobrevin truncation (*Figure 1C* and *Kümmel et al., 2011*); therefore the presence of three charged residues in WT CpxI is expected to strongly disrupt this interaction. Because the observation of heat release in the blocking assays requires the polybasic juxtamembrane region of syntaxin-1 within SNAREΔ60-Sx265, it is most likely that the interaction underlying this heat release involves binding of the

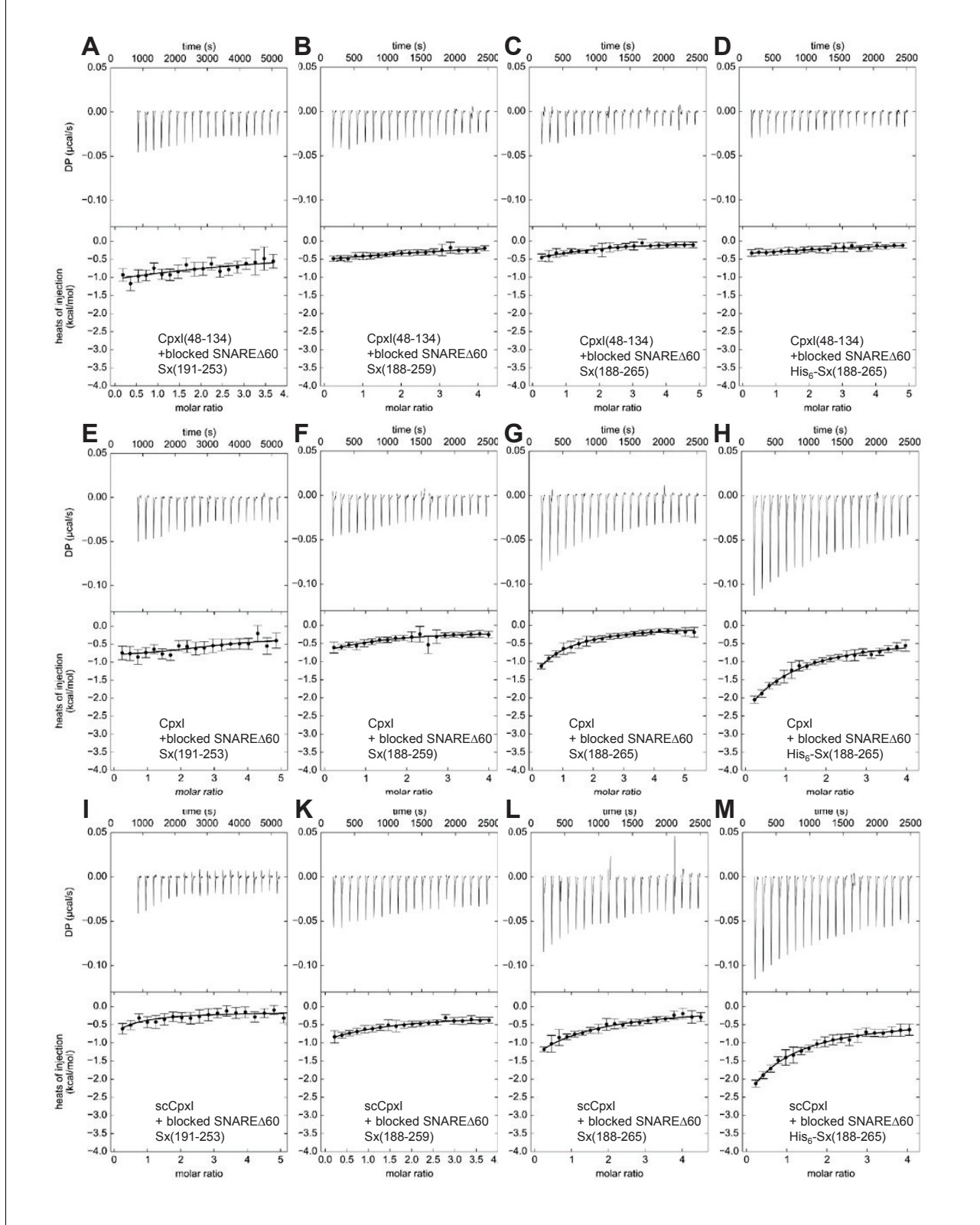

**Figure 3.** ITC analysis of CpxI-SNAREΔ60 interactions through blocking assays. The various panels show blocking assays monitored by ITC where SNAREΔ60 complex blocked with 4.9 equivalents of CpxI(48–134) and containing syntaxin-1(191–253) (**A,E,I**), syntaxin-1(188–259) (**B,F,K**), syntaxin-1(188–265) (**C,G,L**) or His₆-syntaxin-1(188–265) (**D,H,M**) was titrated with CpxI(48–134) itself (**A–D**), CpxI (**E–H**) or scCpxI (**I–M**). The upper panels show the baseline- and singular-value-decomposition-corrected thermograms for the respective experiments. The circles in the lower panels are the integrated

*Figure 3 continued on next page*

*Figure 3 continued*

heats of injection, with the error bars representing estimated errors for these values (*Keller et al., 2012*). The lines in these panels represent the respective fits of the data to a single binding site 'A + B < ->AB' model, but note that no meaningful thermodynamic parameters can be derived from these data sets.

DOI: https://doi.org/10.7554/eLife.30286.004

The following figure supplement is available for figure 3:

**Figure supplement 1.** The superclamp mutation does not alter the heat release observed in the blocking assays.

DOI: https://doi.org/10.7554/eLife.30286.005

juxtamembrane region to acidic side chains within residues 1–47 of CpxI, which include the accessory helix and preceding N-terminal sequence (*Figure 1A*). This type of interaction could occur between CpxI and the blocked SNAREΔ60-Sx265 complex, or between the juxtamembrane region of one SNAREΔ60-Sx265 complex and residues 1–47 of a CpxI molecule that is bound via its central helix to another SNAREΔ60-Sx265 complex. In this 'trans' configuration, CpxI would bridge two SNAREΔ60-Sx265 complexes, which might or might not lead to a zigzag arrangement similar to that observed in the crystal structure of scCpxI(26-83) bound to SNAREΔ60 (note that the SNAREΔ60 complex in the crystal structure did not include the syntaxin-1 juxtamembrane region). Both interactions of CpxI (one involving the central helix and the other involving residues 1–47) could also be established in 'cis' with a single SNAREΔ60-Sx265 complex. In any case, the two interactions do not appear to act cooperatively, as residues 1–47 of CpxI or the syntaxin-1 juxtamembrane region do not markedly increase the affinity of SNAREΔ60 for CpxI (*Figure 2*; *Table 1*). Thus, the heat released by the interaction involving the syntaxin-1 juxtamembrane region with residues 1–47 of CpxI is most likely masked in the direct titrations by the much stronger heat arising from the binding of the CpxI central helix.

The functional significance of the interaction of the syntaxin-1 juxtamembrane region with CpxI might be questioned because it appears to be rather weak, but the interaction could be dramatically enhanced by the high local protein concentrations resulting from localization on a membrane. Indeed, this interaction could underlie a conformational change induced by CpxI in the C-terminus of membrane-anchored SNARE complexes that was recently observed by single-molecule fluorescence resonance energy transfer experiments (*Choi et al., 2016*). However, it is also worth noting that our experiments were performed in solution and, in vivo, the syntaxin-1 juxtamembrane region is expected to interact with negatively charged phospholipids present in the plasma membrane such as PS and PIP$_2$ (*Khuong et al., 2013*). Hence, further research will be required to test whether the interaction of the syntaxin-1 juxtamembrane region with CpxI can occur in the presence of such lipids and whether the interaction is physiologically relevant.

## Materials and methods

### Protein expression and purification

Expression vectors for GST-PreScission human synaptobrevinΔ60 (residues 29–60; SybΔ60), GST-TEV rat syntaxin-1A (residues 191–253), His$_6$-SUMO human SNAP25A-N terminal SNARE motif (residues 7–82; SNAP25N), His$_6$-SUMO human SNAP25A-C terminal SNARE motif (residues 141–203; SNAP25C), His$_6$-SUMO human CpxI (residues 48–134), His$_6$-thrombin human CpxI (residues 1–134), and His$_6$-thrombin human scCpxI (residues 1–134 D27L, E34F, R37A) were described previously by the Rothman laboratory (*Kümmel et al., 2011*). Additionally, vectors for His$_6$-rat syntaxin-1A (residues 188–259) and His$_6$-rat syntaxin-1A (residues 188–265) were also prepared by the Rothman laboratory using standard recombinant DNA techniques. All fusion proteins were expressed in *E. coli* BL21 (DE3) cells by induction with 0.5 mM IPTG at an O.D$_{600}$ of 0.6 for 4 hr at 37°C. Proteins were purified as described (*Kümmel et al., 2011*) and *Trimbuch et al., 2014*). Briefly, cells were harvested and re-suspended in PBS pH 7.4 containing 1 mM TCEP and supplemented with Sigma protease inhibitors. Cleared lysates were applied to either glutathione sepharose resin (GE) or Ni-NTA resin (Thermo Fisher), washed with PBS pH 7.4, and eluted in PBS pH 7.4, 400 mM imidazole. Affinity tags were cleaved with the indicated protease overnight at 4°C. After affinity tag cleavage, all proteins were further purified using size exclusion chromatography on a Superdex S75 column (GE 16/60)

**Table 1.** Summary of $K_D$s (in µM units) between CpxI proteins and SNAREΔ60 complexes containing different syntaxin-1 fragments measured by ITC*.

| | SNAREΔ60-Sx253 | SNAREΔ60-Sx259 | SNAREΔ60-Sx265 | His$_6$-SNAREΔ60-Sx265 |
|---|---|---|---|---|
| CpxI(48-134) | 2.0 [1.4–2.8] | 1.4 [1.2–1.7] | 2.0 [1.6–2.5] | 1.8 [1.5–2.1] |
| CpxI | 1.9 [1.6–2.3] | 2.5 [2.0–3.1] | 2.4 [2.0–3.0] | 2.2 [1.8–2.6] |
| scCpxI | 2.0 [1.8–2.4] | 2.2 [1.7–2.5] | 2.2 [1.9–2.6] | 2.2 [1.9–2.5] |

*At least two independent experiments were performed for each combination of CpxI protein and SNAREΔ60 complex. $K_D$s were derived from global fit of the independent experiments performed for each combination. For all $K_D$s, 68.3% confidence intervals calculated using the error-surface projection method are indicated between brackets.

DOI: https://doi.org/10.7554/eLife.30286.006

equilibrated with 20 mM Tris pH 7.4, 125 mM NaCl, 1 mM TCEP. The expression of Syx 188–265 led to the majority of the protein being expressed in inclusion bodies. Since this fragment does not contain any tertiary structure, a denaturing protocol was used to extract the protein from the pellet after lysis and centrifugation. After extraction in 50 mM Tris pH 7.4, 1M NaCl, and 6 M Gdn-HCl, the protein was applied to Ni-NTA resin, washed with PBS pH 7.4, 1 M NaCl, eluted in PBS pH 7.4, 1 M NaCl, 400 mM imidazole, and dialyzed into buffer containing 20 mM Tris pH 7.4, 1M NaCl. Removal of the affinity tag was performed concomitantly with dialysis for syntaxin-1(188–265), while syntaxin-1(188–265) with an intact His$_6$-tag was immediately flash frozen after elution from the Ni-NTA column in PBS pH 7.4, 1 M NaCl, 400 mM imidazole.

## Isothermal titration calorimetry

ITC experiments were performed using a Microcal ITC200 (Malvern) at 25°C. SNAREΔ60 complexes were prepared by mixing SNAP25N, SNAP25C, SybΔ60 and the corresponding syntaxin-1A fragment in equimolar ratios and incubating overnight at 4°C. Assembled complexes were purified the next day using size exclusion chromatography with a Superdex S75 column (GE 16/60). All proteins were dialyzed (2 L for 4 hr followed by 4 L overnight) in a buffer containing PBS (pH 7.4, 137 mM NaCl, 3 mM KCl, 10 mM phosphate buffer, 0.25 mM TCEP) before the experiments. Protein concentrations were measured by UV absorbance at 280 nm. All experiments were performed at least in duplicate for each combination of CpxI protein and SNAREΔ60 complex to check the reproducibility of the data. For direct titrations (*Figure 2*), CpxI(48-134), CpxI or scCpxI (150 µM) was directly titrated into the chamber containing 8 µM SNAREΔ60-Sx253, SNAREΔ60-Sx259, SNAREΔ60-Sx265 or His$_6$-SNAREΔ60-Sx265. The data were baseline corrected and integrated with NITPIC, fitted with a nonlinear least squares routine using a single-site binding model with ITCsy and plotted with GUSSI (*Brautigam et al., 2016*). The 'A + B < ->AB' model was used for the fitting, and apparent concentration errors for the cell contents were compensated for by refining an incompetent fraction parameter. The 68.3% confidence intervals were obtained using the error surface projection method. Global analysis with ITCsy was performed for each set of experiments carried out with the same protein fragments to derive the $K_D$s described in *Table 1*. For blocking assays, CpxI(48-134), CpxI or scCpxI (300–500 µM) was titrated into the chamber containing 17–21 µM SNAREΔ60-Sx253, SNAREΔ60-Sx259, SNAREΔ60-Sx265 or His$_6$-SNAREΔ60-Sx265 and 4.9 equivalents of CpxI(48-134).

## Acknowledgements

We thank Feng Li, Frederic Pincet, James E Rothman, Karin Reinisch, and Shyam S Krishnakumar for extensive and fruitful discussions that were critical to uncover the importance of the syntaxin-1 juxtamembrane region to observe an interaction between SNAREΔ60 and residues 1–47 of Complexin-I. We also thank Feng Li and Fred Pincet for very insightful comments on the manuscript, and Jeff Coleman for providing the expression vectors for the proteins used in this study, as well as for advice on protein purification. Eric Prinslow was supported by NIH Training Grant T32 GM008297. This work was supported by grant I-1304 from the Welch Foundation (to JR) and by NIH Research Project Award R35 NS097333 (to JR).

## Additional information

### Funding

| Funder | Grant reference number | Author |
|---|---|---|
| National Institutes of Health | R35 NS097333 | Josep Rizo |
| Welch Foundation | I-1304 | Josep Rizo |
| National Institutes of Health | T32 GM008297 | Eric A Prinslow |

The funders had no role in study design, data collection and interpretation, or the decision to submit the work for publication.

### Author contributions

Eric A Prinslow, Chad A Brautigam, Conceptualization, Data curation, Investigation, Methodology, Writing—review and editing; Josep Rizo, Conceptualization, Data curation, Funding acquisition, Investigation, Methodology, Writing—original draft

### Author ORCIDs

Josep Rizo iD http://orcid.org/0000-0003-1773-8311

### Decision letter and Author response

Decision letter https://doi.org/10.7554/eLife.30286.010
Author response https://doi.org/10.7554/eLife.30286.011

## Additional files

### Supplementary files

• Transparent reporting form
DOI: https://doi.org/10.7554/eLife.30286.007

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
