## [Decision Letter]

Thank you for submitting your article "Reconciling Isothermal Titration Calorimetry Analyses of Interactions between Complexin and Truncated SNARE Complexes" for consideration by *eLife*. Your article has been reviewed by three peer reviewers, one of whom is a member of our Board of Reviewing Editors, and the evaluation has been overseen by Randy Schekman as the Senior Editor.

The reviewers have discussed the reviews with one another and the Reviewing Editor has drafted this decision to help you prepare a revised submission.

This work goes a long way to reconcile conflicting findings of two previously published papers (Trimbuch et al. 2014 and Krishnakumar et al. 2015). To recap, Trimbuch et al. could not detect any interaction between the accessory helix of complexin and the ternary SNARE complex (as assessed by ITC and solution NMR), whereas Krishnakumar et al. presented ITC and DLS data of this interaction. In this work, the authors have found that the reason for the discrepancy related to slightly different truncations of syntaxin at the C-terminal end. Upon using a matching construct, the authors now also report an interaction between the complexin accessory helix and the blocked/truncated SNARE complex (Figure 3). Moreover, the authors find that inclusion of a his-tag further enhances this interaction (Figure 3).

Comments:

1) The Kd of the "direct" interactions (Table 1) is around 2.4 microM (Table 1). However, previous ITC experiments by Krishnakumar et al. 2015, reported Kd=457 nM. Is this difference due to the difference in syntaxin constructs used by these two studies, i.e., inclusion of a his-tag by Krishnakumar et al. 2015?

2) While the ITC data in Figure 3 may indeed not produce reliable fits, were attempts made to optimize the concentration in the ITC cell and the syringe in order to obtain better ITC binding curves (similar to those published by Krishnakumar et al. 2015, their Figure 2)?

3) The lack of a statistically significant effect by the superclamp mutation in Figure 3) is perhaps not surprising considering the relatively poor ITC curves. Moreover, the lack of an effect of the superclamp mutant in the direct experiments (Table 1) can of course be explained by the dominant interaction between the core domain of Cpx and the SNARE complex. Thus, the point made about the superclamp mutant should be toned down.

---

## [Author Response]

*This work goes a long way to reconcile conflicting findings of two previously published papers (Trimbuch et al. 2014 and Krishnakumar et al. 2015). To recap, Trimbuch et al. could not detect any interaction between the accessory helix of complexin and the ternary SNARE complex (as assessed by ITC and solution NMR), whereas Krishnakumar et al. presented ITC and DLS data of this interaction. In this work, the authors have found that the reason for the discrepancy related to slightly different truncations of syntaxin at the C-terminal end. Upon using a matching construct, the authors now also report an interaction between the complexin accessory helix and the blocked/truncated SNARE complex (Figure 3). Moreover, the authors find that inclusion of a his-tag further enhances this interaction (Figure 3).*

We thank the reviewers for the nice summary of our paper. We would like to point that we do not really report an interaction between the complexin accessory helix and the blocked/truncated SNARE complex. The interaction revealed by our data involves residue 1-47 of complexin, which include the accessory helix and also the preceding N-terminal region. Hence, it is unclear for the moment whether the accessory helix is involved in the interaction. We did not attempt to address this question because we believe that further studies of this interaction should ideally be performed with new assays involving SNARE complexes anchored to membranes containing PS and PIP_2_.

*Comments:*

*1) The Kd of the "direct" interactions (Table 1) is around 2.4 microM (Table 1). However, previous ITC experiments by Krishnakumar et al. 2015, reported Kd=457 nM. Is this difference due to the difference in syntaxin constructs used by these two studies, i.e., inclusion of a his-tag by Krishnakumar et al. 2015?*

In our hands, the presence of the His-tag did not affect the K_D_s measured in the direct titrations (Table 1, Figure 2). We are not sure what is the reason for the differences in the K_D_s measured in the two labs, but we did not pursue clarification of this apparent discrepancy because the Rothman laboratory told us about important details on the approach used for the blocking assays, i.e. the use of sufficient excess of the CpxI(48-134) blocking fragment so that minimal heat release was observed in control experiments where blocked SNAREΔ60 was titrated with CpxI(48-134) itself (Results and Discussion section). Hence, the (larger) heat release observed in the titrations with CpxI or scCpxI could not arise from incomplete saturation of SNAREΔ60 by CpxI(48-134), and the issue of whether the K_D_ is 457 nM or 2.4 μM was not critical for the interpretation of the blocking assays. In any case, we feel confident about the K_D_ values that we report because of the consistency among the many titrations that we made using various proteins expressed with constructs from the Rothman lab (Figure 2, Table 1), and the similarity of the results to those we obtained previously with proteins made with our own constructs (Trimbuch et al. 2014).

*2) While the ITC data in Figure 3 may indeed not produce reliable fits, were attempts made to optimize the concentration in the ITC cell and the syringe in order to obtain better ITC binding curves (similar to those published by Krishnakumar et al. 2015, their Figure 2)?*

The conditions that we used for the experiments of Figure 3 were similar to those reported in Figure 2 Krishnakumar et al. 2015. The data in this figure cannot be compared with our data presented in Figure 3, as the syntaxin-1 fragment used for the latter did not contain the His_6_-tag and hence leads to lower heat release with a natural increase in noise. The data in Figure 3 (obtained with a syntaxin-1 fragment that did contain the His_6_-tag) may be a little more noisy that those presented in Figure 2 of Krishnakumar et al. 2015, but the superposition of four data sets shown in the new Figure 3—figure supplement 1 (two obtained with WT CpxI and two with scCpxI) shows the consistency of our results. The difficulty in measuring reliable K_D_s that we mention in the paper does not arise from poor quality in our ITC data but from the difficulty in reliably determining the baselines (as we now point out in paragraph five of the Results and Discussion section). This difficulty does not affect the conclusions of the paper and the ability to compare results obtained with WT CpxI and scCpxI (see point 3). We also would like to point out that Krishnakumar et al. 2015 only presented one data set for the blocking experiment with FL CpxI and hence the reliability of the K_D_ that they report for this experiment cannot be assessed.

*3) The lack of a statistically significant effect by the superclamp mutation in Figure 3) is perhaps not surprising considering the relatively poor ITC curves. Moreover, the lack of an effect of the superclamp mutant in the direct experiments (Table 1) can of course be explained by the dominant interaction between the core domain of Cpx and the SNARE complex. Thus, the point made about the superclamp mutant should be toned down.*

We do not consider our ITC curves to be of poor quality. They exhibit the natural noise expected for weak interactions at the protein concentrations used, which were chosen based on the conditions described in Krishnakumar et al. 2015. In our manuscript we did not make any statement about statistically significance with regard to the effect of the superclamp mutation. We stated that the mutation did not markedly alter the heat release observed in the blocking assays and hence that no effect of the mutation can be detected with these assays. This point is now further strengthened by the superpositions of data obtained with WT CpxI and scCpxI shown in Figure 3—figure supplement 1. Even with the natural noise observed in the ITC data, it is clear that the superclamp mutation does not substantially affect the heat release observed in the blocking assays. In this context, we would like to emphasize that the superclamp mutation replaces three charged residues with hydrophobic side chains that pack against the hydrophobic groove left in SNAREΔ60 by the synaptobrevin truncation in the crystal structure of scCpxI(26-83) bound to SNAREΔ60 (Figure 1 and Kummel et al. 2011). Therefore, the presence of three charged residues in WT CpxI is expected to strongly disrupt this interaction, and the heat release observed in the blocking assays is expected to be strongly affected by the mutation if the heat released arises from the interaction of the accessory helix with SNAREΔ60 observed in the zigzag crystal structure. We have modified this part of the Discussion to better explain this point (paragraph five). We agree that the lack of an effect of the superclamp mutation in the direct titrations can be explained by the dominant nature of the interaction involving the central helix and hence we have removed the corresponding sentence in the Discussion.